# Precipitation Downscaling with Spatiotemporal Video Diffusion

**Prakhar Srivastava**[1]    **Ruihan Yang**[1]    **Gavin Kerrigan**[1]    **Gideon Dresdner**[2]
**Jeremy McGibbon**[2]    **Christopher Bretherton**[2]    **Stephan Mandt**[1]
[1]University of California, Irvine    [2]Allen Institute for AI, Seattle
{prakhs2,ruihan.yang,gavin.k,mandt}@uci.edu
{gideond,jeremym,christopherb}@allenai.org

## Abstract

In climate science and meteorology, high-resolution local precipitation (rain and snowfall) predictions are limited by the computational costs of simulation-based methods. Statistical downscaling, or super-resolution, is a common workaround where a low-resolution prediction is improved using statistical approaches. Unlike traditional computer vision tasks, weather and climate applications require capturing the accurate conditional distribution of high-resolution given low-resolution patterns to assure reliable ensemble averages and unbiased estimates of extreme events, such as heavy rain. This work extends recent video diffusion models to precipitation super-resolution, employing a deterministic downscaler followed by a temporally-conditioned diffusion model to capture noise characteristics and high-frequency patterns. We test our approach on FV3GFS output, an established large-scale global atmosphere model, and compare it against six state-of-the-art baselines. Our analysis, capturing CRPS, MSE, precipitation distributions, and qualitative aspects using California and the Himalayas as examples, establishes our method as a new standard for data-driven precipitation downscaling.

## 1 Introduction

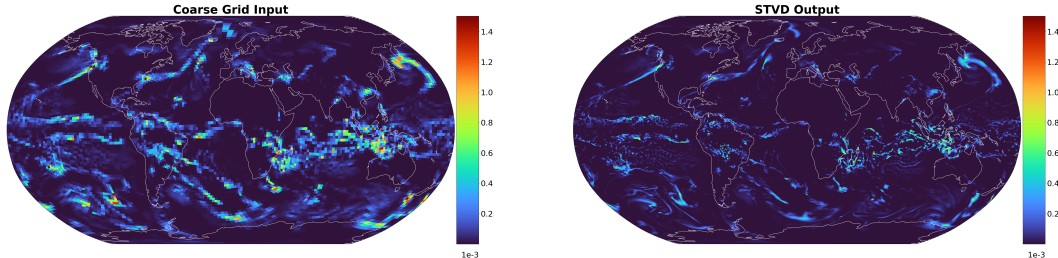

Figure 1: Static snapshot from the Spatiotemporal Video Diffusion (STVD) model, illustrating input (left) and output (right) precipitation frames. The input panel displays simulated coarse-resolution precipitation (rain, snow) fields (Section 3), super-resolved into the high-resolution output shown in the right panel. Both frames use Robinson projection and cover six tiles of the cubed-sphere grid, providing a detailed global view (optimal viewing with zoom). For dynamics, see Fig. 3.

Precipitation patterns are central to human and natural life. In a rapidly warming climate, reliable simulations of changing precipitation patterns can help adapt to climate change. However, these simulations are challenging due to the multi-scale variability of weather systems and the influence of complex surface features (like mountains and coastlines) on precipitation trends and extremes

38th Conference on Neural Information Processing Systems (NeurIPS 2024).

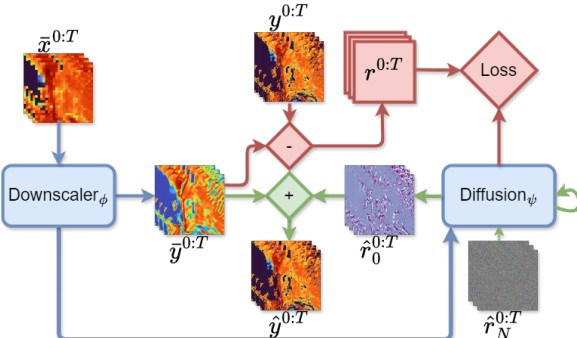

Figure 2: Our model's training and inference pipelines: Blue blocks apply to both phases, red blocks to training only, and green blocks to inference only. It deterministically downscales a low-resolution precipitation sequence using spatio-temporal factorized attention and models residuals with conditional diffusion (with factorized attention). Here, $T$ denotes sequence length and $N$ denotes diffusion steps. The parameters ($\theta = \phi, \psi$) are optimized jointly during training. See A.1 for details.

[58]. For many purposes, such as estimating flood hazards, precipitation must be estimated at spatial resolutions of only a few kilometers. Fluid-dynamical models of the global atmosphere are too expensive to run routinely at such fine scales [65], so the climate adaptation community relies on "downscaling"[1] of coarse-grid simulations to a finer grid. Traditional downscaling methods are either dynamical (running a fine-grid fluid-dynamical model limited to the region of interest, which requires specialized knowledge and computational resources) or statistical (typically restricted to simple univariate methods) [67]. Our work builds on vision based super-resolution methods to improve statistical downscaling and is a natural follow-up to recent deep-learning-based weather/climate prediction methods, which have revolutionized data-driven forecasting. These approaches boast improvements of orders of magnitude in runtime without sacrificing accuracy [46, 29].

We address the downscaling problem for a sequence. Our objective is to transform a sequence ("video") of low-resolution precipitation frames into a sequence of high-resolution frames. Despite differences from natural videos, precipitation's hourly temporal continuity allows us to use video super-resolution techniques to leverage multiple context frames for stochastic downscaling [53, 38].

Recent efforts to enhance the resolution of climate states like precipitation have relied on deterministic regression methods using convolutions or transformers. However, super-resolution is a one-to-many mapping with a continuum of "correct" answers. Supervised learning for these problems often leads to visual artifacts from *mode averaging*, where the network predicts an average of incompatible solutions, causing blurriness in visual data [30, 76]. Besides visual artifacts, mode averaging can have even more dramatic implications in climate and weather modeling, such as the underestimation of extreme precipitation [44], which is mainly induced by regional weather patterns on the unresolved scale. A natural alternative to supervised super-resolution methods [12, 27, 75, 11, 23] to prevent mode averaging is conditional *generative modeling*, which captures multimodal conditional distributions.

To that end, recent works propose using generative adversarial networks (GANs) for precipitation downscaling. These methods often face challenges, tending to converge on specific modes of the data distribution and occasionally fixating on isolated points in extreme cases. Despite their perceptual appeal, the scientific utility of super-resolution requires accurate modeling of the statistical *distribution* of high-resolution data given low-resolution input, which GANs typically fail to capture.

We propose SpatioTemporal Video Diffusion (STVD)[2] for precipitation downscaling. We use a deterministic regression model ("downscaler") for a coarse prediction, refined by a conditional video diffusion model that captures the residual error for adding fine-grained details. Both modules rely on spatio-temporal factorized attention to process the input sequence. Diffusion models are well-suited for precipitation downscaling as they successfully capture high dimensional and multimodal distributions, alleviating a key drawback of GAN-based methods for climate science applications.

---

[1]This is the climate science terminology for super-resolution.
[2]Code : `https://github.com/mandt-lab/STVD`

This study highlights the capability of conditional diffusion models to meet the specific needs of statistical precipitation downscaling, with our key contributions being:

1. We introduce a novel framework for temporal precipitation downscaling using diffusion models. Our model combines a deterministic downscaling module with a diffusion-based residual module. It leverages spatio-temporal factorized attention to process information from multiple low-resolution frames.

2. Our model outperforms six strong super-resolution baselines across multiple criteria, including MSE and several distributional metrics. We compare against two image super-resolution models and four video super-resolution models using the FV3GFS global atmosphere simulation dataset [77, 10].

3. Our approach captures key characteristics of precipitation, including extreme precipitation probabilities and spatial patterns of annual precipitation in mountainous regions, which are crucial for domain science applications.

Our paper is structured as follows: we first describe our method (Sec. 2), followed by our experimental findings (Sec. 3). Finally, we discuss relevant literature (Sec. 4) and its connection to our work. Fig. 1 shows a global view of the input and predicted precipitation of our model. The code for our model is available at `https://github.com/mandt-lab/STVD`.

## 2 Downscaling via Spatiotemporal Video Diffusion

**Problem Statement**    At training time, we assume access to a collection of high-resolution precipitation frame sequences $\mathbf{y}^{0:T}$ and their corresponding low-resolution precipitation frame sequences $\mathbf{x}^{0:T}$. Such a low-resolution sequence can be obtained through area-weighted coarsening [39] of the corresponding high-resolution sequence. The dataset is discussed extensively in Sec. 3. Frame indices are represented by superscripts, where we assume that each sequence consists of $T+1$ frames for simplicity. While it may be possible to roll out predictions for multiple sequences autoregressively using techniques such as reconstruction guidance [21], we leave this exploration for future work. Our objective is to train a model to effectively *downscale*, or *super-resolve* a given sequence $\mathbf{x}^{0:T}$ with $\mathbf{y}^{0:T}$ serving as the target. We use "downscaling" and "super-resolution" interchangeably.

More formally, let $\mathbf{x}^t \in \mathbb{R}^{C \times H \times W}$ and $\mathbf{y}^t \in \mathbb{R}^{1 \times sH \times sW}$ represent individual low-resolution and high-resolution frames. Here, $s \in \mathbb{N}$ denotes the downscaling factor, $C$ is the number of channels (quantities used as input to the model to characterize the atmospheric state in each low-resolution grid cell so as to add skill to the precipitation prediction), and $H, W$ indicate the height and width of the low-resolution frame. For our study, we adopt a downscaling factor of $s = 8$ and have $C = 12$ total low-resolution channels. In addition to the low-resolution precipitation state, we provide eleven channels of information to the model, such as topography, wind velocity, and surface temperature; see A.2 for details.

**Solution Sketch**    Our approach treats the downscaling problem as a conditional generative modeling task. We devise a model to learn the conditional distribution of high-resolution precipitation frames, incorporating contextual information from the low-resolution precipitation frame sequence.

Our proposed solution, **SpatioTemporal Video Diffusion (STVD)** (Fig. 2), relies on two modules: a deterministic downscaler and a stochastic component based on conditional diffusion models [20, 63], both using spatio-temporal factorized attention. The first module uses a UNet with factorized attention to integrate information from a low-resolution frame sequence, resulting in an initial prediction frame sequence $\bar{\mathbf{y}}^{0:T}$. The second module is a conditional diffusion model that stochastically generates a sequence of additive residual frames $\mathbf{r}^{0:T}$ which serves to add fine-grained details to the initial prediction. Together, these two modules produce a high-resolution frame sequence $\hat{\mathbf{y}}^{0:T} = \bar{\mathbf{y}}^{0:T} + \mathbf{r}^{0:T}$. Both modules are trained end-to-end.

Decomposing the prediction into a deterministic mean and a stochastic residual is inspired by predictive-coding-based video decompression. This approach aims to predict a sequence of video frames while compressing the sparse residuals [2, 74] which are easier to model than dense frames. Similarly, it is easier to generate residuals than dense images when using diffusion models [73, 41].

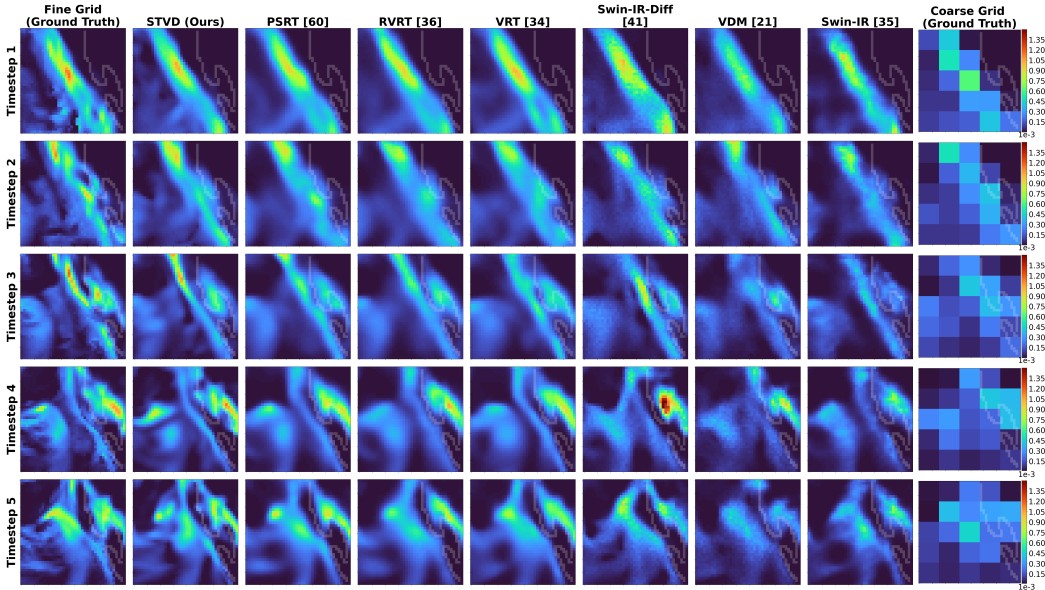

Figure 3: A qualitative comparison between our proposed model and top baseline for a precipitation event associated with a cold front impinging on the Northern California coast and then the Sierra mountain range (coastline marked in hazy white). Fig. 6 plots the regional topography. The time interval between adjacent frames is 3 hours; the plotted region is $1000 \times 1000$ km. Our model resolves the fine-grid precipitation structure better than the considered baselines. See A.3 for full-page high quality samples from Himalayas and Sierra.

In what follows, we first describe our overall probabilistic framework for downscaling. Then, we discuss the deterministic module along with spatio-temporal attention, followed by the remaining residual prediction module based on diffusion generative modeling. See A.1 for architecture details.

## 2.1 Probabilistic Modeling of Downscaling

Given a sequence of low-resolution frames $\mathbf{x}^{0:T}$ and the corresponding high-resolution frames $\mathbf{y}^{0:T}$, we aim to learn a parametric approximation $p_\theta$ of the conditional distribution $p\left(\mathbf{y}^{0:T} \mid \mathbf{x}^{0:T}\right) \approx p_\theta\left(\mathbf{y}^{0:T} \mid \mathbf{x}^{0:T}\right)$. Importantly, we do not assume independence across time; each generated frame $\mathbf{y}^t$ can depend on all other generated frames. The generated high-resolution frame sequence is conditioned on the entire low-resolution frame sequence, capturing long-range temporal correlations and enhancing the fidelity and cohesion of the high-resolution reconstruction.

As noted earlier, the likelihood $p_\theta\left(\mathbf{y}^{0:T} \mid \mathbf{x}^{0:T}\right)$ is modeled using a deterministic downscaler and a residual diffusion model. We will discuss how the model parameters $\theta = (\phi, \psi)$ decompose into those for a downscaler ($\phi$) and a diffusion model ($\psi$).

### 2.1.1 Deterministic Downscaling

Our first module is a deterministic downscaler that predicts an initial high-resolution frame sequence $\bar{\mathbf{y}}^{0:T} = \mu_\phi\left(\mathbf{x}^{0:T}\right)$ where $\mu_\phi$ is a network generating a deterministic high-resolution prediction with parameters $\phi$. We perform bicubic interpolation on each frame of $\mathbf{x}^{0:T}$ before passing the sequence through the network $\mu_\phi$. Since the diffusion network operates on high-resolution inputs (i.e. denoising the high-resolution residuals), this choice allows us to use the same UNet [52] architecture (with different weights) for both the downscaling module $\mu_\phi$ and the residual diffusion module. This enables us to easily share features across the modules via concatenation. See A.1 for further details.

Importantly, $\mu_\phi$ incorporates a temporal attention mechanism that allows any frame at time $t$, or its corresponding feature map, to attend to all context frames from 0 to $T$. This architecture enables the concurrent inference of all frames within the sequence $\bar{\mathbf{y}}^{0:T}$. The attention weights differ for each frame, allowing for the flexible incorporation of information across time.

### 2.1.2 Stochastic Residual Modeling via Diffusion

After computing the initial prediction $\bar{\mathbf{y}}^{0:T}$, finer details are modeled by residuals learned from a conditional diffusion model. Our final stochastic high-resolution frame sequence $\hat{\mathbf{y}}^{0:T}$ is generated by sampling an additive residual sequence $\mathbf{r}^{0:T}$ from this model: $\hat{\mathbf{y}}^{0:T} = \bar{\mathbf{y}}^{0:T} + \mathbf{r}^{0:T}$. Thus, we seek to model the residuals $\mathbf{r}^{0:T} = \mathbf{y}^{0:T} - \bar{\mathbf{y}}^{0:T}$. Our diffusion model generates the entire residual sequence $\mathbf{r}^{0:T}$ concurrently, with the generation of each residual $\mathbf{r}^t$ dependent on the others. This is achieved via a UNet architecture with spatio-temporal attention, similar to the mechanism used for the deterministic downscaling module. See A.1 for further details.

To model the distribution of $\mathbf{r}^{0:T}$, we use DDPM [20]. To that end, we introduce a collection of latent variables $\mathbf{r}^{0:T}_{0:N}$, where the lower subscripts indicate the denoising diffusion step. In the *forward process*, the latent variable $\mathbf{r}^{0:T}_n$ is created from $\mathbf{r}^{0:T}_{n-1}$ via additive noise. In the *reverse process* for generation, a denoising model (with parameters $\psi$) is trained to predict $\mathbf{r}^{0:T}_{j-1}$ from $\mathbf{r}^{0:T}_j$. $N$ denotes the total number of denoising steps. Note that $\mathbf{r}^{0:T} = \mathbf{r}^{0:T}_0$, i.e. the first diffusion step corresponds to the true residual. Additionally, $\mathbf{r}^{0:T}_0$ implicitly depends on the downscaler parameters $\phi$, allowing us to simultaneously optimize all model parameters $\theta = (\phi, \psi)$ within the context of diffusion modeling.

As is standard in diffusion models [20], we parameterize the reverse process via a Gaussian distribution with a mean determined by a neural network $M_\psi$,

$$p_\psi\left(\mathbf{r}^{0:T}_{n-1}|\mathbf{r}^{0:T}_n, \mathbf{c}\right) = \mathcal{N}\left(\mathbf{r}^{0:T}_{n-1}|M_\psi\left(\mathbf{r}^{0:T}_n, n, \mathbf{c}\right), \gamma\mathbf{I}\right), \tag{1}$$

where $M_\psi$ is a denoising network and $\gamma$ is a hyperparameter for variance. The diffusion model directly accesses the context $\mathbf{c} = (\mathbf{x}^{0:T}, \bar{\mathbf{y}}^{0:T})$, and is implicitly conditioned on $\mathbf{x}^{0:T}$ via concatenation of feature maps from the downscaler module. As in the downscaler, we bicubically upsample $\mathbf{x}^{0:T}$ before channel-wise concatenation with $\bar{\mathbf{y}}^{0:T}$ to match the dimensions when forming $\mathbf{c}$.

### 2.1.3 Loss Function

To train our model, we use the angular parametrization suggested by [59]. Specifically, this results in the diffusion loss of the form

$$L\left(\psi, \phi\right) = \mathbb{E}_{\mathbf{x}^{0:T}, \mathbf{y}^{0:T}, n, \epsilon} \sum_{t=0}^{T} \left\|\mathbf{v} - M_\psi\left(\mathbf{r}^{0:T}_n, n, \mathbf{c}\right)\right\|^2 \tag{2}$$

where $\epsilon \sim \mathcal{N}(0, I)$, $n$ is sampled uniformly from $\{1, 2, \ldots, N\}$, and the sequences $\mathbf{x}^{0:T}$, $\mathbf{y}^{0:T}$ are sampled from the training distribution. Here, $\mathbf{c} = (\mathbf{x}^{0:T}, \bar{\mathbf{y}}^{0:T})$ where $\bar{\mathbf{y}}^{0:T} = \mu_\phi\left(\mathbf{x}^{0:T}\right)$. The scalars $\alpha_n^2 = \prod_{i=1}^n (1 - \beta_i)$ and $\sigma_n^2 = 1 - \alpha_n^2$ are used to define $\mathbf{v} \equiv \alpha_n\epsilon - \sigma_n\mathbf{r}^{0:T}_0$. Training and inference are concurrent across multiple frames due to spatio-temporal attention. Alg. 1 and 2 demonstrate the training and sampling strategy under the angular parametrization. We use DDIM sampling [62] to generate frame residuals with fewer diffusion steps.

### 2.1.4 Network Architecture

Both the downscaler and the conditional diffusion model employ a UNet backbone with similar architectures and key adaptations to the attention mechanism (see A.1). The downscaler takes the multi-channel input frames $(\mathbf{x}^{0:T})$, yielding an initial estimate $(\bar{\mathbf{y}}^{0:T})$. The diffusion UNet conditions on diffusion step $n$ and concatenates feature maps from the downscaler with its own. The concatenated input to the diffusion UNet ($\mathbf{x}^{0:T}$, $\bar{\mathbf{y}}^{0:T}$, and $\mathbf{r}^{0:T}_n$), along with the conditioning variables (diffusion step $n$ and the feature maps from downscaler), yields the output $\mathbf{v}$.

Computing full attention for temporal coherence across the entire video data cube is very expensive for processing long sequences or high-resolution inputs. To optimize efficiency, we decouple attention between spatial and temporal dimensions, use a linear variant of self-attention [26] for non-bottleneck layers (where the effective number of "tokens" for attention is relatively large), focus spatial attention on localized patches (instead of the entire feature map, which could be wasteful), and calculate per-channel temporal attention in large spatial dimensions (namely, the ultimate and penultimate expansion and contraction layers of UNet). These modifications dramatically reduce the time complexity and memory footprint of these transformer blocks.

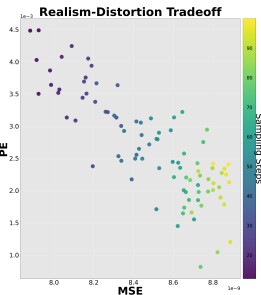

Figure 4: Tradeoff between mean square error and percentile error (see Sec. 3). Inference at Himalayan region (see Figs. 6 and 12).

| **Algorithm 1:** Training STVD | **Algorithm 2:** Sampling STVD |
|---|---|
| **while** *not converged* **do** | Get an equally spaced increasing sub-sequence $\tau$ of length $K \ll N$; |
| $\quad$ Sample $\mathbf{x}^{0:T}$ and $\mathbf{y}^{0:T}$; | $\bar{\mathbf{y}}^{0:T} = \mu_\phi \left( \mathbf{x}^{0:T} \right)$; |
| $\quad n \sim \mathcal{U}(0, 1, 2, .., N)$; | $\mathbf{c} = \left( \mathbf{x}^{0:T}, \overline{y}^{0:T} \right)$; |
| $\quad \epsilon \sim \mathcal{N}(\mathbf{0}, \mathbf{I})$; | $\mathbf{r}_K^{0:T} \sim \mathcal{N}(\mathbf{0}, \mathbf{I})$; |
| $\quad \bar{\mathbf{y}}^{0:T} = \mu_\phi \left( \mathbf{x}^{0:T} \right)$; | **for** *n in reversed($\tau$)* **do** |
| $\quad \mathbf{r}_0^{0:T} = \mathbf{y}^{0:T} - \bar{\mathbf{y}}^{0:T}$; | $\quad \hat{\mathbf{v}} = M_\psi \left( \mathbf{r}_n^{0:T}, n, \mathbf{c} \right)$; |
| $\quad \mathbf{v} = \alpha_n \epsilon - \sigma_n \mathbf{r}_0^{0:T}$; | $\quad \hat{\mathbf{r}} = \alpha_n \mathbf{r}_n^{0:T} - \sigma_n \hat{\mathbf{v}}$; |
| $\quad \mathbf{r}_n^{0:T} = \alpha_n \mathbf{r}_0^{0:T} + \sigma_n \epsilon$; | $\quad \hat{\epsilon} = \frac{\sigma_n}{\alpha_n} \left( \mathbf{r}_n^{0:T} - \hat{\mathbf{r}} \right)$; |
| $\quad \mathbf{c} = \left( \mathbf{x}^{0:T}, \overline{y}^{0:T} \right)$; | $\quad \mathbf{r}_{n-1}^{0:T} = \alpha_{n-1} \hat{\mathbf{r}} + \sigma_{n-1} \hat{\epsilon}$; |
| $\quad \hat{\mathbf{v}} = M_\psi \left( \mathbf{r}_n^{0:T}, n, \mathbf{c} \right)$; | $\hat{\mathbf{y}}^{0:T} = \bar{\mathbf{y}}^{0:T} + \mathbf{r}_0^{0:T}$; |
| $\quad L = ||\mathbf{v} - \hat{\mathbf{v}}||^2$; | |
| $\quad (\psi, \phi) = (\psi, \phi) - \nabla_{\psi,\phi} L$; | |

## 3 Experiments

We conduct a comprehensive evaluation of our proposed method, SpatioTemporal Video Diffusion (STVD), against six contemporary state-of-the-art baselines. The first two baselines are image super-resolution models based on the Swin Vision Transformer (Swin-IR)[35] and its residual diffusion variant (Swin-IR-Diff). The next two baselines are video super-resolution models grounded in vision transformer architecture (VRT) [34] and its recurrent variant (RVRT) [36]. The latter incorporates guided deformable attention for clip alignment, enhancing its temporal modeling capabilities. We compare against another video-super-resolution baseline (PSRT) [60] which also relies on the transformer architecture but uses multi-frame attention groups. Finally, we compare against a video diffusion baseline (VDM) [21]. Fig. 1 shows a global view of the input and predicted precipitation.

We perform ablation studies in three configurations. In the first two, we experiment with the input sequence length. While our proposed model uses a context length of 5 frames, we also conduct experiments with 3 frames and 1 frame (STVD-3 and STVD-1). Note that using a single context frame ablates for the temporal attention block as well. The third ablation (STVD-Single) involves removing the additional input channels (i.e. only providing the model with the low-resolution precipitation sequence) to assess their impact on performance metrics. In summary, our experiments demonstrate that our method outperforms all baselines across all metrics considered. Additionally, our ablation studies highlight the importance of temporal context and additional climate inputs.

**Dataset** Our dataset derives from an 11-member initial condition ensemble of 13-month simulations using a global atmosphere model, FV3GFS, run at 25 km resolution and forced by climatological sea surface temperatures and sea ice. The first month of each simulation is discarded to allow the simulations to spin up and meteorologically diverge, effectively providing 11 years of reference data (of which first 10 years are used for training and the last year for validation). FV3GFS, developed by the National Oceanic and Atmospheric Administration (NOAA), is a version of NOAA's operational global weather forecast model ([77, 10]).

Three-hourly average data were saved from this entire simulation, which used a 25 km horizontal "fine grid". We further coarsened the selected fields by a factor of 8 to create a 200 km horizontal "coarse grid", resulting in paired data $(x_t, y_t)$, where $x_t$ is the coarse-grid global state and $y_t$ is the corresponding fine-grid global state. Our goal is to apply video downscaling to the coarse-grid precipitation field to obtain temporally smooth fine-grid precipitation estimates that are statistically similar to the true data. This approach is attractive because many fine-grid precipitation features, such as cold fronts and tropical cyclones, are poorly resolved on the coarse grid but are temporally coherent across periods much longer than 3 hours. We use 12 coarse-grid input fields, including precipitation, topography, and horizontal vector wind at various levels. See A.2 for the list of included atmospheric variables. FV3GFS uses a cubed-sphere grid, where the surface of the globe is divided into six tiles, each of which is covered by an $S \times S$ array of points. Our data fields reflect this structure with $S = 48$ for the 200 km coarse grid and $S = 384$ for the 25 km fine grid.

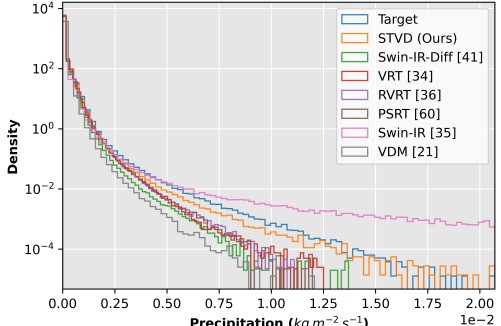

Figure 5: Distributions of the fine-grid three-hourly average precipitation, for all gridpoints around the globe. The Swin-IR baseline over-estimates large precipitation events, whereas all other baselines underestimate key extreme and rare precipitation events. Our model aligns best with the fine-grid ground truth than any the other model. This is also evident with the the EMD and PE metrics discussed in Tab. 1 and Sec. 3.

Table 1: Quantitative comparison between our method and other competitive baselines. EMD represents the Earth-Mover Distance, PE denotes the 99.999th percentile error and SAE is the spatial-autocorrelation error. Overall, our proposed method (STVD) outperforms the baselines across all metrics. In our ablation study, the exclusion of additional side information (STVD-single) or decrement in context length (STVD-3 and STVD-1) appreciably degrades performance.

| | CRPS $(10^{-5})$ | MSE $(10^{-8})$ | EMD $(10^{-6})$ | PE $(10^{-3})$ | SAE $(10^{-6})$ |
|---|---|---|---|---|---|
| **STVD (ours)** | **1.85** | **0.59** | **2.49** | **1.2** | **4.00** |
| PSRT [60] | 2.15 | 0.66 | 4.21 | 3.8 | 6.24 |
| RVRT [36] | 3.55 | 1.73 | 4.33 | 3.6 | 7.39 |
| VRT [34] | 3.58 | 1.74 | 4.61 | 4.0 | 7.39 |
| Swin-IR-Diff [41] | 2.29 | 1.94 | 6.38 | 4.4 | 7.70 |
| VDM [21] | 2.21 | 0.73 | 12.70 | 6.4 | 8.84 |
| Swin-IR [35] | 2.36 | 2.29 | 17.40 | 23.40 | 18.9 |
| STVD-single | 1.81 | 0.62 | 4.64 | 2.3 | 6.09 |
| STVD-3 | 1.96 | 0.68 | 4.94 | 2.6 | 4.99 |
| STVD-1 | 2.05 | 0.72 | 7.19 | 4.1 | 6.87 |

The application presented here serves as a pilot for broader uses of our methodology. Fine-grid simulations are significantly more computationally expensive than coarse-grid simulations (an 8-fold reduction in grid spacing requires almost 1000x more computation), so a coarse-grid simulation with super-resolved details in desired regions could be highly cost-effective for many applications.

During training, our model randomly selects data from one of the six tiles. This strategy ensures that the model learns from the diverse spatial contexts and weather regimes that produce precipitation worldwide. Post-training, for localized analysis, we selectively sample super-resolved precipitation channels from regions with complex terrain, such as California (Fig. 6). These regions can systematically pattern the precipitation on fine scales. This analysis helps us to see how well the super-resolution can learn the time-mean spatial patterns (e.g. precipitation enhancement on the windward side of mountain ranges and lee rain shadows) in the fine-grid reference data.

**Training and Testing Details** We downscale a sequence of precipitation frames from FV3GFS output by a factor of 8. Our approach (STVD) trains on 5 consecutive frames that are downscaled jointly. We optimize our model end-to-end with a single diffusion loss using Adam [28] with an initial learning rate of $1 \times 10^{-4}$, decaying to $5 \times 10^{-7}$ with cosine annealing during training, executed on an NVidia RTX A6000 GPU. The diffusion model is trained using v-parametrization [59], with a fixed diffusion depth (N = 1400). Random tiles extracted from the cube-sphere representation of Earth, with dimensions 384 in high-resolution and 48 in low-resolution, are used during training. We train for one million steps, requiring approximately 7 days on a single node (slightly less for ablations). We use a batch size of one, apply a logarithmic transformation to precipitation states, and normalize to the range $[-1, 1]$. During testing, we employ DDIM sampling with 30 steps on an Exponential Moving Average (EMA) variant of our model (for full frame size), with a decay rate of 0.995.

**Baseline Models** We compare our generative setup against several recent high-performing transformer-based video super-resolution models. These models are trained deterministically. The first, Video Restoration Transformer (VRT) [34], allows for parallel frame prediction and long-range temporal dependency modeling. The second, recurrent VRT (RVRT) [36], incorporates guided deformable attention for effective clip alignment, enhancing its temporal modeling capabilities. The third, PSRT [60], removes the alignment module and modifies the attention window. We also compare against the recent Video Diffusion Model (VDM) [21] which employs global quadratic attention.

To assess the benefits of multi-frame downscaling, we compare with Swin-IR [35], a popular image super-resolution model that harnesses Swin Transformer blocks. However, Swin-IR is trained in a supervised fashion. Thus, as a generative baseline, we compare to Swin-IR-Diff. This model generates a deterministic prediction using Swin-IR [35], followed by modeling a stochastic residual

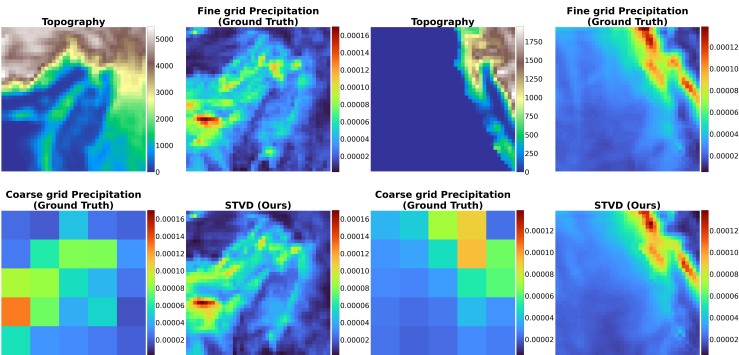

Figure 6: Precipitation over two regions (left: Himalayas; right: Northern California coast, same region as Fig. 3), averaged across a year, for our STVD model and the ground-truth. For each half, the topography of the region is shown in the corresponding top-left whereas the predicted annual average is shown in the corresponding bottom-right. Annually-averaged precipitation is an important indicator of water availability in a region. STVD successfully captures many details of the precipitation that are tied to local topography and are too fine to be resolved the coarse-grid data.

using diffusion. This baseline is inspired by concurrent work on single-image radar-reflectivity downscaling [41], where a UNet is used instead of Swin-IR. See A.2 for details.

**Evaluation Metrics** We evaluate our model differently from standard vision tasks. In addition to the Mean Square Error (MSE), which measures the average squared difference between predicted and actual values but lacks full distributional information, we use several distribution-level metrics for a more meaningful comparison. One such metric is the Continuous Ranked Probability Score (CRPS) [8, 66], which assesses the discrepancy between the predicted cumulative distribution function and the observed data. We compute CRPS over 10 stochastic realizations of our predictions. Fig. 7 visualizes several of these samples.

Furthermore, given the distinctive light-tailed exponential distribution of the precipitation climate state, it is crucial to ensure that downscaling does not significantly alter the distribution of precipitation rates. This necessitates two additional metrics. First, we compute the Earth Mover (or 1-Wasserstein) Distance [54] to quantify the agreement between the target and predicted global precipitation distributions, which are strongly affected by high-resolution details. Second, we focus on tail events and extreme precipitation by considering the 99.999th percentile error (PE), providing a nuanced understanding of the model's performance on rare and extreme precipitation events.

To further assess the spatial fidelity of our downscaling approach, we use the Spatial Autocorrelation Error (SAE) [68]. This metric calculates the mean absolute error between the spatial autocorrelation of the predictions and ground truth. Low SAE ensures that the spatial patterns and the fine structure in precipitation data are preserved during downscaling, which is critical for accurate climate modeling.

**Qualitative and Quantitative Analysis** Tab. 1 provides a quantitative evaluation comparing our method with state-of-the-art baselines and ablations. Our model (STVD) performs strongly across all metrics, outperforming all baselines. We highlight the distributional characteristics in Fig. 5. Swin-IR overestimates precipitation, while all other baselines underestimate it. This discrepancy is undesirable, as poor performance on rare and extreme precipitation events can negatively imapct disaster mitigation policies. In contrast, our method closely matches the precipitation distribution, as measured by PE and EMD.

Using only precipitation as an input (STVD-single) results in slightly worse performance across all metrics, indicating the predictive value of additional inputs. In contrast, our ablation model STVD-1, which lacks full sequence information, performs significantly worse, highlighting the importance of temporal attention in our approach (which decays as a function of time lag as shown in Fig. 14).

Figs. 3, 11 and 12 depict the performance of our model compared to other baselines on examples of a precipitation feature interacting with mountainous terrain. Our model generates high quality results

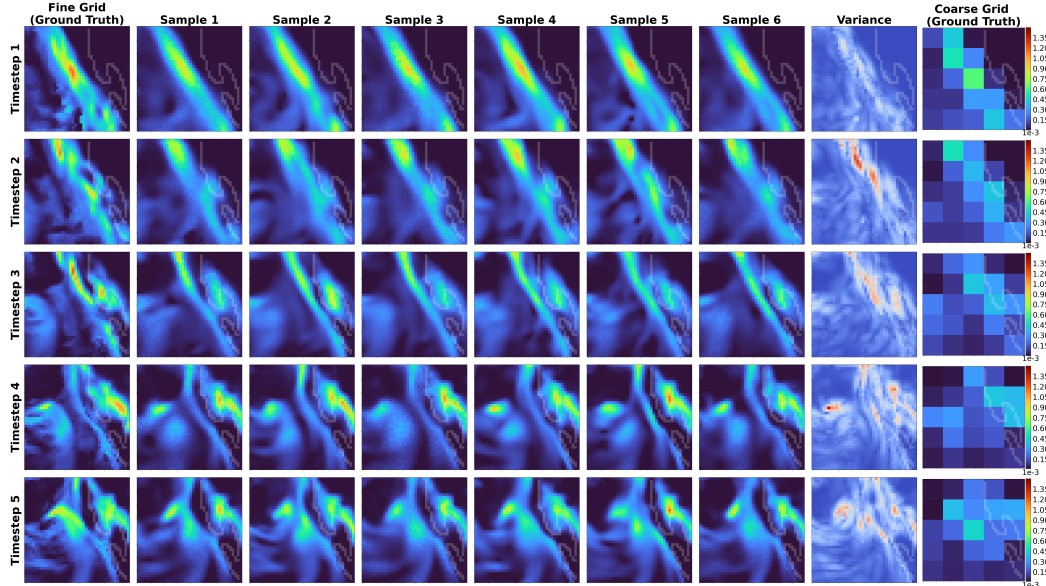

Figure 7: A visualization of the stochastic samples predicted by STVD for a given coarse-grid data. The precipitation event is the same as depicted in Fig. 3. Additionally, we also plot a variance map over the set of these samples to analyze the stochasticity better. Red regions correspond to high variance whereas blue regions correspond to low variance. Model stochasticity seems to be meaningful since the variance is large where mean precipitation is large.

which preserves most patterns with a high degree of similarity. PSRT and RVRT produce slightly more diffuse precipitation features, while Swin-IR produces slightly more pixelated features.

Fig. 6 shows annually-averaged precipitation from the patches in Figs. 3 and 12. Accurately capturing the fine-grid structure of time-mean precipitation is crucial for assessing long-term water availability. Our method (which includes fine-grid topography as a training input) effectively replicates the ground truth. This includes the strength and narrow spatial structure of high precipitation bands along the Northern California coastal mountains and the Sierras. These features are not resolved by the coarse-grid inputs to the super-resolution. See A.3 for full-page high-resolution samples and spectral analysis. Fig. 13 reveals that the spectra for baselines decay more rapidly than for STVD.

**Realism-Distortion Tradeoff**  Distortion metrics such as MSE often conflict with perceptual quality, where reducing distortion typically degrades perceptual realism [7]. In our context, this tradeoff translates to balancing MSE and PE. While MSE captures the average accuracy of predictions, PE represents the model's ability to reproduce extreme events, thereby serving as a proxy for realism. Realism in climate modeling refers to the accurate representation of extreme weather patterns, which are crucial for applications like flood forecasting and disaster mitigation. PE is a distributional criterion that effectively captures these tail events, offering a robust measure of realism. Fig. 4 illustrates this tradeoff, with darker colors corresponding to fewer STVD sampling steps. As the number of diffusion sampling steps increases, MSE tends to rise slightly, but PE decreases significantly. Depending on the application, this tradeoff may potentially be exploited by practitioners. Essentially, the conditional mean minimizes MSE, so any deviation from it increases MSE—even if the deviation appears more realistic. As for sampling steps, fewer steps correspond to larger time increments in the diffusion process. At one extreme, a single step predicts the conditional mean, minimizing MSE. Conversely, more sampling steps more accurately simulate diffusion, generating diverse, realistic samples that increase MSE while reducing PE.

## 4   Related Work

**Diffusion Models**  Diffusion models [61, 20, 64, 43, 45, 40] are a class of generative models based on an iterative denoising process. Closely related to our work are diffusion models for video. Recent

models [73] generate deterministic next-frame predictions autoregressively with additional residuals generated by a diffusion model, or generate videos directly in pixel space [18, 69, 21] or in a latent space [6, 5]. While some works on video diffusion [6, 21] employ video super-resolution as a step in the overall modeling process, our work focuses exclusively on the video super-resolution task, particularly within the context of precipitation downscaling.

**Super-Resolution**   Within the computer vision community, the paradigm for single image super-resolution has shifted from classical approaches [4, 13] to deep learning based methods [71]. Generative approaches, like cascaded diffusion [49, 55], SR3 [56], and DiffPIR [78] employ diffusion models for image super-resolution. However, these are unable to leverage temporal context. On the other hand, many approaches for video super-resolution have been proposed [9, 14, 22]. For a more comprehensive overview, see [38]. Recent models of note include the transformer-based models PSRT [60] and VRT [34], as well as the recurrent variant RVRT [37], which focuses on parallel decoding and guided clip alignment. We emphasize that these state-of-the-art approaches are deterministic, where our approach is generative. This allows us to preventing mode averaging and to produce more realistic samples, which is particularly critical in the context of precipitation modeling.

**Data-driven weather and climate modeling**   Recent years have seen advancements in data-driven climate and weather modeling [51, 42], with models like GraphCast [29], GenCast [48], and Four-CastNet [46] providing forecasts that are competitive with meteorological methods while being significantly faster. Rather than replacing numerical forecasting methods, our approach seeks to augment their capabilities by downscaling coarse-grid predictions.

While downscaling for climate and weather has been approached using techniques based on domain knowledge [25, 24], we focus here on data-driven approaches. [68] draw inspiration from FRVSR [57], adopting an iterative approach that uses the high-resolution frame estimated in the previous step as input for subsequent iterations. [72] employed Fourier neural operators for downscaling at arbitrary resolutions. [16] generate physically consistent downscaled climate states, using a softmax layer to enforce conservation laws. These approaches, though, are deterministic and trained by minimizing the MSE, thus lacking the realism and uncertainty quantification provided by a generative approach.

In terms of generative approaches, concurrent work [41] employs diffusion models for downscaling climate states. The use of GANs has also been pervasive in downscaling and precipitation prediction [32, 47, 17, 50, 15, 70]. However, these GAN-based approaches inherit the mode collapse and training difficulties present in all GAN-based models [3]. Here, we highlight that these approaches are applied at each frame and cannot incorporate temporal information as is done in our model.

Beyond downscaling, [1] demonstrate the diffusion model's efficacy in synthesizing full rain density from vorticity inputs. Additionally, [19] uses a diffusion model for downscaling solar irradiance. These models are also used for probabilistic weather forecasting and nowcasting [31, 33].

## 5   Conclusion

We propose a video super-resolution method for probabilistic precipitation downscaling. Our model, STVD, deterministically super-resolves a given low-resolution frame sequence and then stochastically models the residual details via diffusion. Our model effectively resolves how fine-grid precipitation features, generated as weather systems, interact with complex topography based on temporally coherent coarse-grid information. Our method outperforms several competitive baselines on a range of quantitative metrics. This is an important step towards designing effective statistical downscaling methods, providing highly localized information for planning against extreme weather events, such as floods or hurricanes in a warming climate, using tractable coarse-grid atmospheric models.

**Limitations and Broader Impacts**   A limitation of our approach is the necessity of paired low-resolution and high-resolution images for training. While this can be done once prior to training, designing methods that only require the low-resolution states is an interesting challenge. In terms of broader impacts, our approach could potentially have harmful consequences if adopted blindly to a new dataset, where distribution shift could cause the model performance to degrade, potentially leading to underestimation of extreme weather risks such as droughts or floods. To mitigate these risks, the model should be re-trained and rigorously evaluated on the dataset of interest.

## Acknowledgments and Disclosure of Funding

We thank Kushagra Pandey and Yibo Yang for valuable feedback. Prakhar Srivastava was supported by the Allen Institute for AI summer internship for much of this work. Stephan Mandt acknowledges support from the National Science Foundation (NSF) under an NSF CAREER Award IIS-2047418 and IIS-2007719, the NSF LEAP Center, by the Department of Energy under grant DE-SC0022331, the IARPA WRIVA program, the Hasso Plattner Research Center at UCI, and by gifts from Qualcomm and Disney. Gavin Kerrigan is supported in part by the HPI Research Center in Machine Learning and Data Science at UC Irvine.

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

# Precipitation Downscaling
# with Spatiotemporal Video Diffusion

*Supplementary Materials*

## A.1  Model Architecture

Our architecture is a conditional extension of the DDPM [20] and SR3 [56] models. As discussed in
Sec. 2, Fig. 8 presents the architecture of the proposed *denoising* and *downscaling* networks, while
Fig. 9 provides detailed description of each component. Before elaborating on the specifics, we define
the naming conventions for the parameter choices adopted in this section:

- `ChannelDim`: `ResBlock` channel dimension for first contractive layer of UNet.
- `ChannelMultipliers`: channel dimension multipliers for subsequent contractive layers
  (including the first layer) in both the downscaling and denoising modules. The expansive
  layer multipliers follow in reverse.
- `ResBlock`: a standard ResBlock implementation consisting of two blocks, each with a
  weight-standardised convolution using a $3 \times 3$ kernel, Group Normalization over groups of
  8 and SiLU activation, followed by a channel-adjusting $1 \times 1$ convolution.
- `Attention`: a standard implementation of quadratic or linear attention, incorporating 4
  attention heads, each with a 32-dimensional representation. Using a $3 \times 3$ convolutional
  kernel, query, key, and value feature maps are generated, resulting in feature maps of size [B
  $\times$ T, H $\times$ C, X, Y], where B, T, C, X, and Y denote batch, time, channel, height, and width,
  respectively, with H representing the number of heads. These feature maps are rearranged
  to [BATCH, HEAD, CHANNEL, TOKEN]), facilitating self-attention between TOKENs of
  CHANNEL dimensions. The rearrangement order determines whether spatial or temporal
  self-attention is performed. Subsequently, the feature maps revert to their original format,
  upon which a separate convolution operation is conducted, followed by layer normalization
  to project the feature maps back to their original dimensions, akin to their state before the
  initial convolution within the attention block. Further elaboration on the rearrangement
  choices for the variants is discussed, adhering to *einops*[3] notation:
  - `Q-Spatial`: quadratic variant applied in the bottleneck layer; self attends between
    every pixel of the feature map with the following rearrangement : [B $\times$ T, H $\times$ C, X,
    Y] -> [B $\times$ T, H, C, X $\times$ Y]
  - `Q-Temporal`: quadratic variant applied in the bottleneck layer; self attends between
    feature maps across time with the following rearrangement : [B $\times$ T, H $\times$ C, X, Y] ->
    [B, H, C $\times$ X $\times$ Y, T]
  - `L-Spatial`: linear variant applied in expansive and contractive layers of UNet; self
    attends between every pixel within a patch of the feature map with the following
    rearrangement : [B $\times$ T, H $\times$ C, X $\times$ P, Y $\times$ P] -> [B $\times$ T, H $\times$ X $\times$ Y, C, P $\times$ P] where
    P is patch size, starting with 192, halving at each contractive layer and doubling at each
    expansive layer.
  - `L-Temporal`: linear variant applied in expansive and contractive layers of UNet; self
    attends between feature maps across time in a channel factorised manner with the
    following rearrangement : [B $\times$ T, H $\times$ C, X, Y] -> [B, H $\times$ X $\times$ Y, C, T]
- `MLP`: conditioning on the denoising step $n$ is achieved through this block, which uses 32-
  dimensional random Fourier features, followed by a linear layer, GELU activation and
  another linear layer to transform the noise step to a higher dimension.
- `Cov/TransCov`: these are convolutional ($3 \times 3$ kernel) downsampling and upsampling
  blocks that change the spatial size by a factor of 2.

Fig. 8 illustrates the interaction between the U-Net architecture of the denoising and downscaling
networks. The top U-Net depicts the denoising network, while the bottom U-Net depicts the

---

[3]`https://einops.rocks/1-einops-basics`

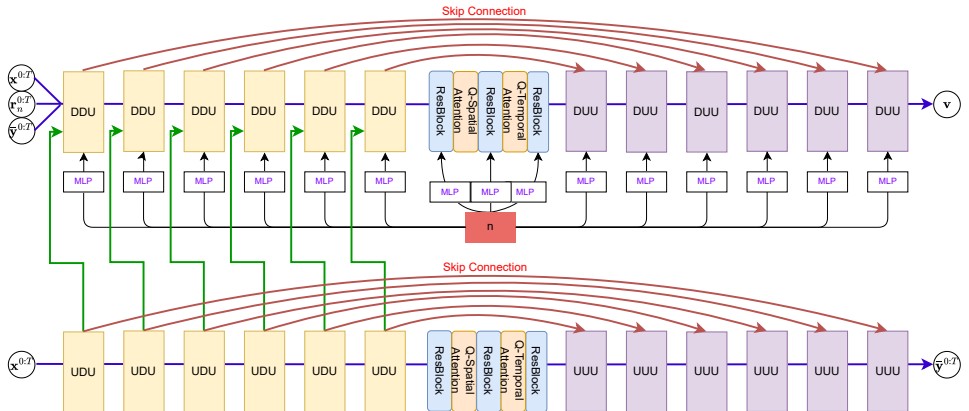

Figure 8: The figure depicts the overall model architecture where the top UNet performs diffusion on the residual, conditioned on the noise step $n$, $\mathbf{x}^{0:T}$, $\bar{\mathbf{y}}^{0:T}$ and the context of the bottom U-Net feature map (refer to Eq. (1) in Sec. 2.1). The bottom UNet is the deterministic downscaler. Details of each block are shown in Figure 9.

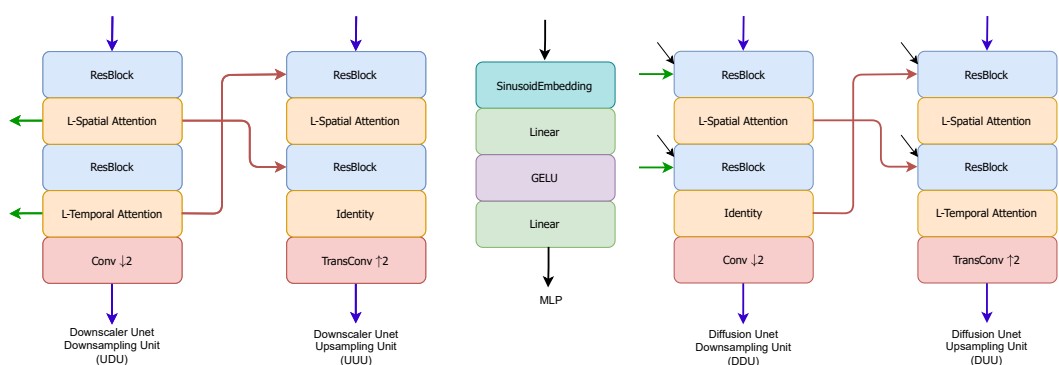

Figure 9: The details of the components of the modules shown in Figure 8; the colored arrows in the modules correspond to the arrows with the same color in Figure 8.

downscaling network. The denoising network is conditioned in three ways. First, the network is conditioned on the bicubically downscaled low-resolution frames $\mathbf{x}^{0:T}$, concatenated with both the noisy residual $\mathbf{r}_n^{0:T}$ the downscaler output and $\bar{\mathbf{y}}^{0:T}$ along the channel dimension. Second, the network is also conditioned on the feature maps generated by the downscaler network as indicated by the green arrows connecting the downsampling units of both the networks. The L-(Spatial/Temporal) Attention blocks from contractive layers of the downscaler network yield a feature map that gets concatenated with the inputs of both ResBlocks of the contractive layer of the denoising network. Finally, each contractive and expansive layer of the diffusion UNet gets conditioned on the denoising step $n$, shown via the **black** arrows. This conditional embedding for the step is generated through MLP and received by both ResBlocks. The information flows from the noisy residual through the network as shown by the blue arrows to predict the angular parameter $\mathbf{v}$. Both U-Nets have skip connections, indicated by the red arrows, between both ResBlocks of contractive and expansive layers of the same UNet.

Table 2: FV3GFS uses a cubed-sphere grid, in which the surface of the globe is divided into six tiles. Each high-resolution tile covers 25 km and is $384 \times 384$. Our UNet Encoder and Decoder have 6 layers with a base channel dimension of 64 and multipliers as stated above.

| TileSize | ChannelDim | ChannelMultipliers |
|---|---|---|
| $384 \times 384$ | 64 | 1,1,2,2,3,4 |

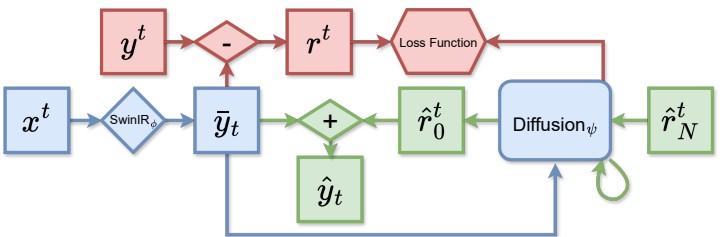

Figure 10: An illustration of the training and inference pipelines of Swin-IR-Diff. Similar to Fig. 2, blue blocks represent operations common to both training and inference phases. Red blocks signify operations exclusive to training, while green blocks indicate inference-only processes. However, in contrast, it is an image-only downscaler. This model takes in a current low-resolution frame which is deterministically downscaled via Swin-IR, followed by modeling of the residual details via conditional diffusion. The model details remain similar to what is described in A.1 with the absence of downscaler-conditioning and temporal attention in the diffusion model.

Table 3: Additional Variables in FV3GFS dataset.

| Short Name | Long Name | Units |
|---|---|---|
| CPRATsfc | Surface convective precip. rate | $kg/m^2/s$ |
| DSWRFtoa | Top of atmos. down shortwave flux | $W/m^2$ |
| TMPsfc | Surface temperature | K |
| UGRD10m | 10-meter eastward wind | m/s |
| VGRD10m | 10-meter northward wind | m/s |
| ps | Surface pressure | Pa |
| u700 | 700-mb eastward wind | m/s |
| v700 | 700-mb northward wind | m/s |
| liq_wat | Vert. integral of cloud water mix ratio | $kg/kg\ kg/m^2$ |
| sphum | Vert. integral of specific humidity | $kg/kg\ kg/m^2$ |
| zsurf | Topography | – |

## A.2 On Swin-IR-Diff and Multiple Channels

Here, we discuss SwinIR-Diff, which expands on one of our robust baselines. Sec. 3 provides a concise overview of Swin-IR-Diff. Shown as a sketch in Fig. 10, this model opts to downscale each precipitation state individually, akin to an image super-resolution model. Resembling SR3 in its foundation of a conditional diffusion model, Swin-IR-Diff adopts a residual pipeline. It involves a deterministic prediction corrected by a residual generated from the conditional diffusion model, with the Swin-IR model serving as the deterministic downscaler in this context.

We conducted an ablation, focusing on the incorporation of additional climate states as input to our precipitation downscaling model STVD. The rationale for including these states is drawn from the insights of [17], who justified a similar selection for the task of precipitation forecast based on domain science. While Tab. 3 provides detailed information on the various states employed, the utility of these states is closely examined in Tab. 1, with specific attention to STVD (multiple input states) and STVD-single (only precipitation state as input). Clearly, the introduction of additional channels yields a notable improvement in performance.

## A.3 Additional Samples

In addition to re-illustrating precipitation downscaling in the Sierras and Central California from Fig. 3 in Fig. 11), we present our model's output for another unique region—the Himalayas. Fig. 12 mirrors Fig. 11, displaying outputs from different models. Note that for the same regional topography, Fig. 6 (left) compares the annual precipitation time average. We also provide video samples corresponding to Fig. 11 (`california.gif`) and Fig. 12 (`himalaya.gif`) in the supplementary zip file.

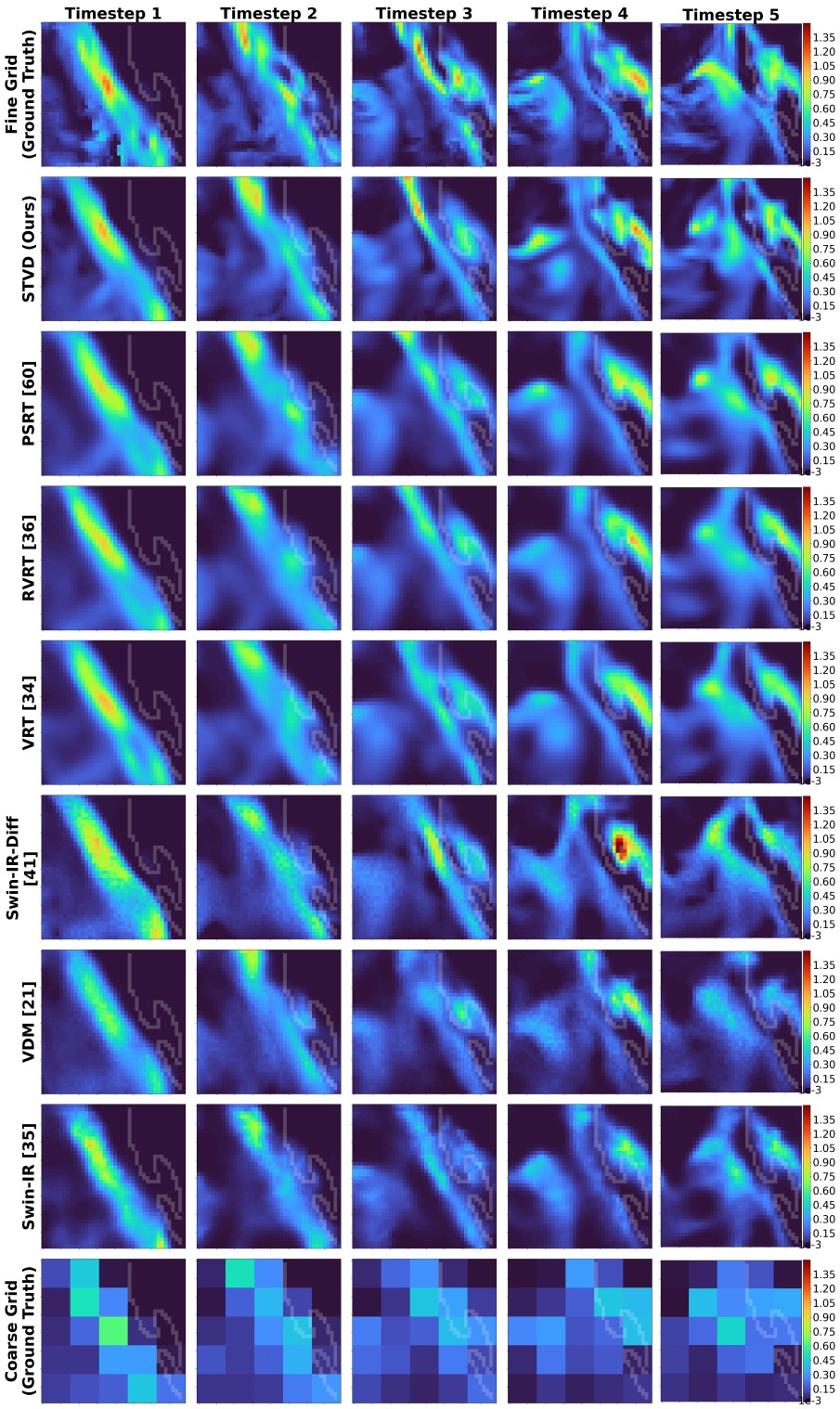

Figure 11: Qualitative comparison between our proposed model and all baselines for a specific precipitation event in the Sierra mountain range. This figure is a repetition of Fig. 3 for better visual overview. The first row represents the ground truth fine-grid precipitation state sequence, and the last row represents the coarse-grid precipitation that is being downscaled. All other rows correspond to our model and the baseline outputs. The time interval between adjacent frames is 3 hours; the plotted region is $1000 \times 1000$ km.

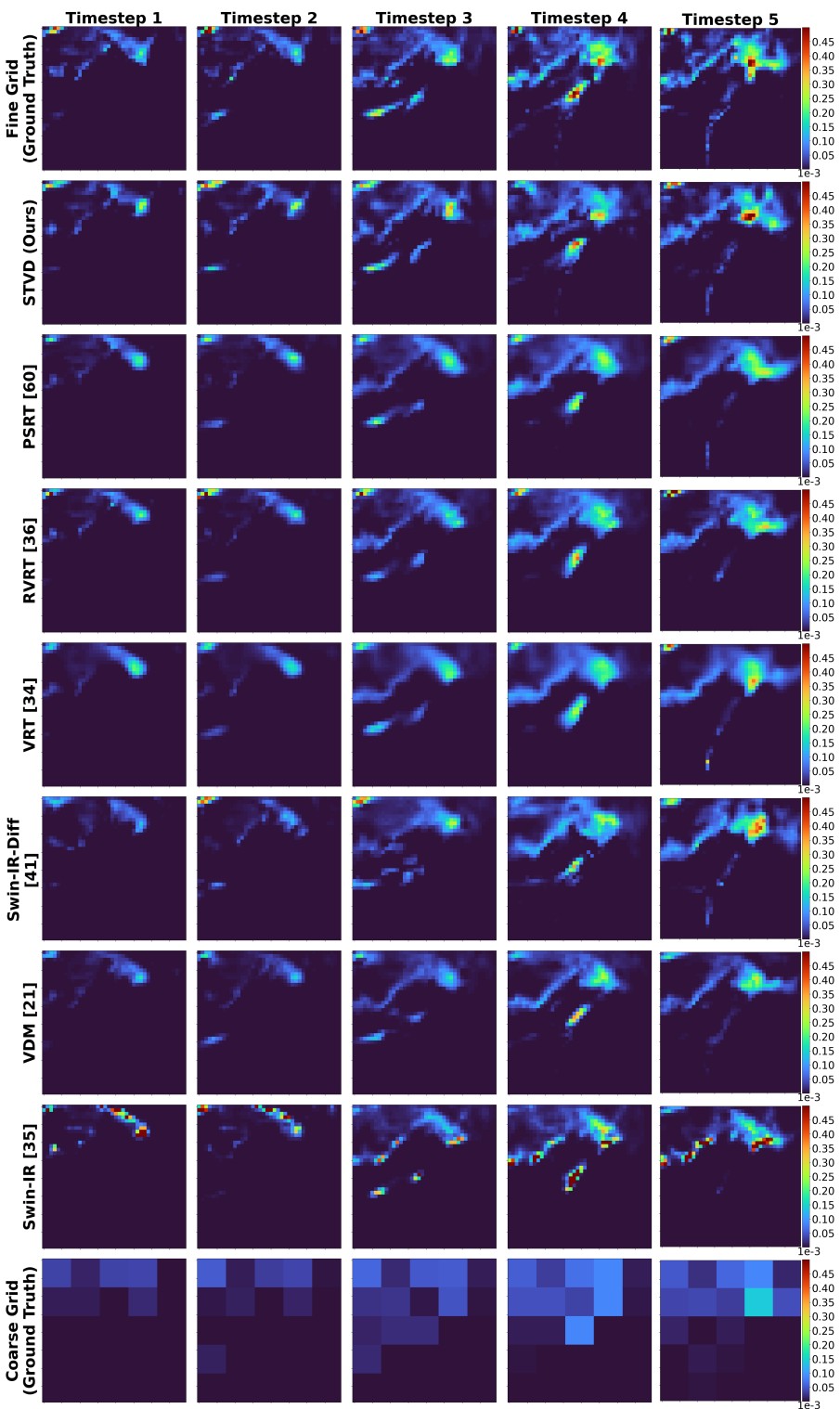

Figure 12: Another qualitative comparison between our proposed model and baselines for a specific precipitation event in the Himalayan mountain range. Fig. 6 (left) plots the regional topography. Similar to Fig. 3, the first row represents the ground truth fine-grid precipitation state sequence, and the last row represents the coarse-grid precipitation that is being downscaled. All other rows correspond to our model and the baseline outputs. The time interval between adjacent frames is 3 hours; the plotted region is $1000 \times 1000$ km.

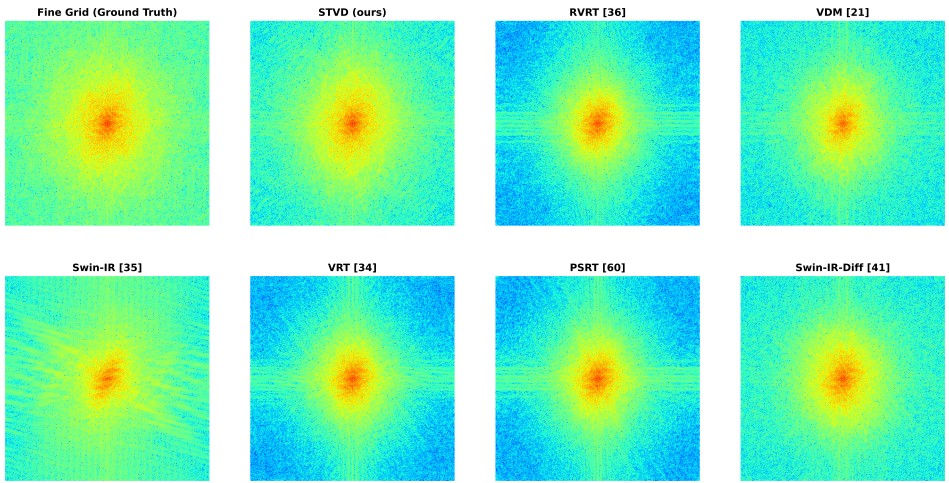

Figure 13: RVRT, VRT, PSRT and Swin-IR-Diff show a spectra which decays too rapidly, i.e. placing too little energy on the high-frequency components. The Swin-IR baseline exhibits artifacts in the spectra. Overall, the spectra of samples from our method most closely match the ground-truth spectra.

### A.3.1 Spectra

In Fig. 13, we plot (in log-scale) the squared-magnitude of the complex-valued FFT applied to an image in our evaluation set. Overall, we see that the samples from STVD closely match the ground-truth high resolution spectrum. The baselines RVRT, VSRT and PSRT demonstrate a spectrum which decays too rapidly, placing too little energy in the high-frequency components. Additionally, we see a banding in these spectra. This indicates that these baselines are overly smooth compared to the ground-truth and follow similar banding patterns of the corresponding low resolution image. For the Swin-IR baseline, we observed outliers of large magnitudes in the generated precipitation maps, which we hypothesize leads to the observed checkerboard pattern seen in the spectrum. Swin-IR-Diff and VDM seem to decay the spectra more rapidly than STVD.

### A.3.2 Temporal Attention Behaviour

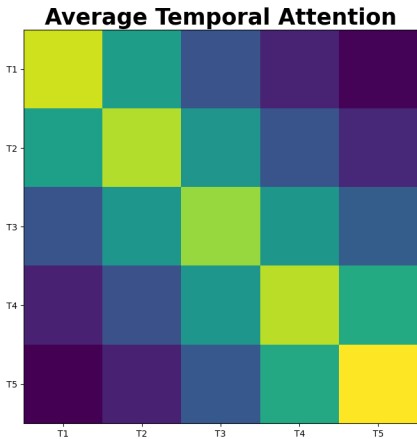

Figure 14: A visualization of the temporal attention weights averaged over the entire validation set and attention heads for the bottleneck layer of the deterministic downscaler. $T1 - T5$ denotes the temporal sequence. The weights evidently decay as a function of temporal distance which makes physical sense. For example, feature map at position T2 attends the most towards itself along with immediate temporal neighbors at T1 and T3. Lighter colors correspond to larger weight.

