# OpenReview forum: "Precipitation Downscaling with Spatiotemporal Video Diffusion"
_NeurIPS.cc/2024/Conference — NeurIPS 2024 poster_

### Official Review · Reviewer_giVy · 2024-07-05

**Soundness:** 2
**Presentation:** 3
**Contribution:** 2
**Rating:** 5
**Confidence:** 5

**Summary:**

This paper presents a novel framework for spatio-temporal precipitation downscaling, comprising two modules: a deterministic downscaling module and a diffusion module. The model is able to outperform the SOTA models, especially in extreme events and in mountainous areas.

**Strengths:**

1. Multiple losses, such as PE and EMD, are used to measure the effectiveness of the results. At the same time, the discussion of the trade-off between realism and bias is novel and informative, using MSE to represent the average accuracy of predictions and PE to represent the model's ability to reproduce extreme events.
2. The model outperforms six strong super-resolution baselines and can be established as a new standard for data-driven precipitation downscaling.

**Weaknesses:**

Experiments are insufficient. Such as the effectiveness of the sharing features across the modules is not proven by ablation experiments.

**Questions:**

1. In line 121, “bicubic interpolation” does not have a explanation. Can you explain it?
2. In line 121, what is ”pixelated features” ?
3. Is the input, additional climate states, the L1 data or L2 data?

**Limitations:**

Is this work of practical value? The authors acknowledge that switching to a different dataset requires retraining due to differences in data distribution. However, can the model be trained if high-resolution ground truth images are not available? If high-resolution images are available, does it make sense to reduce their resolution before training the model?

---

> ### Author Rebuttal · Authors · 2024-08-07
>
> _Thank you for taking the time to read our work. We are happy you find novelty in the realism-distortion tradeoff and appreciate the PE and EMD metrics. We address your concerns as follows:_
>
> - ___“...the effectiveness of the sharing features across the modules is not proven by ablation experiments.”:___
>
>     We thank the reviewer for suggesting an additional ablation study. Due to time constraints, we are unable to run this ablation for the rebuttal, but we will include this ablation in an updated version of our paper. However, we feel that this design choice is a relatively minor part of our overall architecture, and our existing experiments demonstrate the strength of our proposed method (reviewer EgC8) and ablate the key components of our model (reviewer St3G). We would be glad to discuss any additional experiments the reviewer feels would further strengthen our submission.
>
> - ___Explanation of "bicubic interpolation":___
>
>     Bicubic interpolation is a resampling method that uses the values of the 16 nearest pixels (4x4 grid) to estimate the value of each pixel in the upsampled image. This method ensures a smoother and higher-quality image than simpler methods like nearest-neighbor or bilinear interpolation. In our context, since the input and output dimensions of our UNet are the same, we use bicubic interpolation to upsample the low-resolution input to match the high-resolution output dimensions (scale factor of 8). We will include this explanation in the revised manuscript for clarity.
>
> - ___Explanation of "pixelated features":___
>
>     By "pixelated features," we refer to traditional pixelation artifacts or block artifacts that can occur in super-resolution tasks. These artifacts are characterized by a blocky appearance in the upscaled image, where individual pixels or groups of pixels become visibly distinct, leading to a loss of detail and smoothness. We will clarify this in the manuscript.
>
> - ___“Is the input, additional climate states, the L1 data or L2 data?”:___
>
>     We appreciate the reviewer's question about the input data and additional climate states. However, we are unsure about the specific reference to "L1 data" or "L2 data" in this context. Could you please clarify what you mean by these terms? In our study, the input includes both the primary low-resolution precipitation data and additional climate states. These additional covariates are provided at a low resolution. Refer to the STVD-single experiment to highlight the importance of this side data. Additionally, we clarify in the appendix how these side data were selected.
>
> - ___“Is this work of practical value?...”:___
>
>     As discussed in [1], fluid-dynamical emulators of the global atmosphere are too expensive to run routinely at such fine scales. So the climate adaptation community relies on “downscaling” of coarse-grid simulations to a finer grid. Our work builds on vision-based super-resolution methods to improve statistical downscaling by allowing a cheap run of fluid-dynamics emulators on a coarse grid followed by a downscaling using our model on the region of interest. Our model's practical value lies in its ability to provide quick and cheap high-resolution downscaling for regions of interest. Additionally, our model emulates long-term annual trends pretty well, which is critical for applications like water availability and management. The main motivation behind this project was to create computationally efficient and realistic proxies for climate emulators:
>     - Generate high-resolution training data for a limited time horizon (e.g., one year).
>     - Train the super-resolution model using this data.
>     - Generate 'cheap' climate predictions at low resolution and use the super-resolution model for focussed regional predictions.
>
>     Please also refer to our global response for a discussion of this point.
>
>     We will also add a discussion in the paper revision on the general issue of data distribution shifts and the potential for developing domain adaptation techniques for climate applications, highlighting it as an interesting future research direction. This context will underscore the practical value and broader applicability of our approach.
>
> _Again, we appreciate the time you took reading our work. We will integrate our responses, along with additional clarifications, into the paper. These enhancements will contribute to the overall readability, clarity and completeness._
>
> _[1] Stevens, B., Satoh, M., Auger, L., Biercamp, J., Bretherton, C. S., Chen, X., ... & Zhou, L. (2019). DYAMOND: the DYnamics of the Atmospheric general circulation Modeled On Non-hydrostatic Domains. Progress in Earth and Planetary Science, 6(1), 1-17._

---

> > ### Comment · Reviewer_giVy · 2024-08-12
> >
> > Thanks to the authors for the detailed response. I acknowledge I have read the rebuttal.

---

### Official Review · Reviewer_St3G · 2024-07-07

**Soundness:** 3
**Presentation:** 3
**Contribution:** 3
**Rating:** 7
**Confidence:** 5

**Summary:**

The method proposes a diffusion model to statistical downscale precipitation. The model requires a combination of high resolution and low resolution video data for training (a common scenario in weather and climate modeling). The diffusion model takes in a video of low resolution atmospheric variables and outputs a sample video of high-resolution precipitation.

In experiments, the model is trained on an ensemble of year-long runs from the standard US forecasting system (FV3GFS). Comparisons are made to deterministic downscaling models. The models are evaluated using mean squared error (MSE), continuous rank probability score (CRPS), earth mover distance, 99.999th percentile error (PE), and spatial autocorrelation error (SAE). The results confirm the advantage of using a probabilistic method in estimating the risk of extreme events.

**Strengths:**

- The use of machine learning to downscale atmospheric variables, particularly precipitation, is a hot topic in weather forecasting and climate modeling. This paper will be of interest to the community.
- Nice ablation studies showing the importance of the temporal dimension and additional atmospheric variables.

**Weaknesses:**

- There have been a number of papers using diffusion models for downscaling (Hatanaka, et al. 2023 for solar irradiance) and nowcasting (Gao, et al. 2023 and Yu, et al. 2024 for precipitation). These are missing from the related work section.
- The use of transformer-based models is not justified. It seems likely that vision transformer models have a poor inductive bias for this task, which probably has a lower degree of long-range spatiotemporal dependency than in natural images/video. The authors mention "key adaptations to the attention mechanism" but these appear to be simplifications that reduce computation and increase the locality bias --- have the authors tried removing attention all together?
- Some typos that could be caught with a spell checker.
- The baselines used for comparison are weak. They are all deterministic, so it is expected that they will fail to capture the extremes. I don't think it is necessary to have comparisons, but I think this point could be made more clearly in the text.
- In experiments, only 10 samples are taken from the diffusion model. This seems small in the context of estimating the risk of extreme events.
- Some clarifications would be helpful (see questions below).

**Questions:**

- 280: For an annual average, the diffusion model isn't really necessary. Wouldn't any of the deterministic statistical downscaling models would perform just as well? This should be clarified.
- 294: It was unclear from the text why additional sampling steps in the STVD increases MSE. I imagine this is because the deterministic downscaling model predicts the mean of the possible rainfall (low MSE) and sampling from the distribution of residuals with the diffusion model will almost certainly increase this. So more sampling steps means a better sample of the residual, which means a higher MSE?

**Limitations:**

- As the authors note, the model is demonstrated on simulation data where high-resolution video inputs are provided. In many applications of statistical downscaling, the input and output are from different sources (e.g. observed high-res vs. GFS model output or reanalysis), and there may be inconsistencies. So the experiments done here are on "clean" data, and further experiments are needed to evaluate the method on applications where it will be useful.

---

> ### Author Rebuttal · Authors · 2024-08-07
>
> _Thank you for taking the time to read our manuscript. We are happy you see the utility of our work in climate modeling and appreciate the ablation study. We address your concerns as follows:_
> - ___Missing related work:___
>
>     We appreciate the reviewer's suggestion, and will include these references. Hatanaka et al. focus on solar irradiance, and Gao et al. and Yu et al. focus on nowcasting. Our work addresses the challenges posed by the temporal dynamics in precipitation downscaling by leveraging temporal attention mechanisms to encourage consistency over time. We emphasize that downscaling fundamentally differs from nowcasting, as we are provided the entire low-resolution video when doing downscaling and thus do not predict future frames.
> - ___“use of transformer-based models not justified… have the authors tried removing attention?”:___
>
>     Before addressing this, we would like to clarify that our model's backbone is not a Vision Transformer but a Conv UNet with Spatial and Temporal Attention blocks applied to the convolutional features. Removing attention would remove the temporal context, resulting in our “STVD-1” ablation, where the performance degrades appreciably. Although 3D convolutions are an alternative, they are very expensive. To visualize the inner workings of temporal attention, we have added Figure 2 in the attached file in the global response section. It visualizes the temporal attention map from the bottleneck layer of the downscaler. We have averaged this map over the whole validation data and multiple attention heads. As one may expect, we see that, on average, attention decays as a function of time lag, meaning that the model learns to assign more weight to features that are temporally closer.
> - ___Typos:___
>
>     Thank you for pointing this out. We will correct all typos in the final draft.
>
> - ___“comparisons are weak... all deterministic... expected that they will fail to capture the extremes…”:___
>
>      Note that not all baselines we used are deterministic. Swin-IR-Diff and VDM are diffusion-based baselines and thus stochastic. Despite being diffusion-based, our experiments reveal that these methods do not capture extreme events as effectively as our model. As shown in Table 1, our method outperforms these baselines in both the PE and EMD metrics. Furthermore, Figure 4 demonstrates that the precipitation distribution produced by our model closely matches the actual distribution, particularly in the extremes, better than the distributions generated by the baselines. Additionally, we have used more competitive vision VSR baselines, acknowledging that most state-of-the-art baselines for VSR remain deterministic.
>
> - ___“only 10 samples, seems small…”:___
>
>     We may have miscommunicated this. To clarify, the histogram in Figure 4 is calculated across *all grid points* over the entire one year of validation data (~$10^8$ points). The 10 samples referred to in the context are the 10 stochastic samples generated from the same input condition, which are used specifically for calculating CRPS.
>
> - ___“annual average, the diffusion model isn't necessary…”:___
>
>     We stress that predicting annual averages was not our prediction goal but rather serves as a *diagnostic* with high relevance to climate modeling practitioners. The primary use case of our model is to provide high-resolution instantaneous downscaling, which captures detailed local behavior and dynamic precipitation patterns. While deterministic methods may capture broad annual trends, our model simultaneously captures both local behavior and annual patterns, which are crucial for practical applications such as water availability and management.
>
> - ___“unclear why additional sampling steps in the STVD increases MSE…”:___
>
>     Yes, precisely, your interpretation is correct. We agree that this could have been explained more clearly in the submission, and we will include a more detailed discussion in the final draft. To elaborate, the conditional mean is the theoretical minimizer of the MSE, and any deviation from this conditional mean would result in a higher MSE -- even if the resulting deviation is more realistic. Regarding the relationship with sampling steps, fewer sampling steps correspond to taking larger time steps in the diffusion process. At one extreme, taking a single step would correspond to predicting the conditional mean (see [1] for a discussion), minimizing the MSE. Conversely, increasing the number of sampling steps results in a more accurate simulation of the diffusion process, producing more diverse and realistic samples, and thus increasing the MSE while decreasing the PE as shown in Figure 3.
>
> - ___“the experiments done here are on "clean" data, and further experiments are needed…”:___
>
>     As discussed in [2], fluid-dynamical emulators of the global atmosphere are too expensive to run routinely at such fine scales. So the climate adaptation community relies on “downscaling” of coarse-grid simulations to a finer grid. Our work builds on vision-based super-resolution methods to improve statistical downscaling by allowing a cheap run of fluid-dynamics emulators on a coarse grid followed by a downscaling using our model on the region of interest. Please also refer to our global response for a discussion of this point.
>
> _Thank you again for taking the time to read our work, as well as for the many suggestions and typo highlighting. We will incorporate these changes in the paper revision to improve readability, as well as the additional content prompted by your feedback._
>
> _[1] Karras, T., ... & Laine, S. (2022). Elucidating the design space of diffusion-based generative models. Advances in neural information processing systems, 35, 26565-26577._
>
> _[2] Stevens, B., Satoh, M., Auger, L., Biercamp, J., Bretherton, C. S., Chen, X., ... & Zhou, L. (2019). DYAMOND: the DYnamics of the Atmospheric general circulation Modeled On Non-hydrostatic Domains. Progress in Earth and Planetary Science, 6(1), 1-17._

---

> > ### Author Response · Authors · 2024-08-12
> >
> > Hi, we wanted to follow up on our recent rebuttal submission to ensure that our responses adequately addressed your concerns. If there are any remaining issues or if you have any further questions, we would be grateful for your feedback.
> >
> > Thank you for your time and consideration.

---

### Official Review · Reviewer_EgC8 · 2024-07-24

**Soundness:** 3
**Presentation:** 4
**Contribution:** 2
**Rating:** 5
**Confidence:** 2

**Summary:**

This paper extends video diffusion model to precipitation super-resolution, where a deterministic downscaler is used to produce initial results and a temporally-conditioned diffusion model is utilized to refine previous coarse results. By combing deterministic and statistical downscaling models, "mode averaging" problems are obviously alleviated. Experimental results demonstrates its effectiveness.

**Strengths:**

1. The paper is well organized, the motivations and method details are clearly described.
2.  The explanation of professional terms is very good, so that researchers in other fields can easily read the paper.
3. It is very reasonable to predict the low-frequency part with a deterministic model and then generate the high-frequency residual with a statistical model.
4. The comparisons with other methods in experiment parts seems sufficient.

**Weaknesses:**

There are some statements, method and experiment details remain to be clear.
1. Is there any design that guarantees that the output high-resolution frames are smooth over the time series?
2. For the high-frequency prediction part, if given the same conditions but different sampling noise, will output completely different results?
3. The authors state that the generative models can capture multimodal conditional distributions and alleviate underestimation of extreme precipitation. Could the authors show experimentally that their approach is better at modeling extreme precipitation than traditional supervised methods?

**Questions:**

Please refer to Weaknesses part.

**Limitations:**

The authors clearly point out the limitations of their work and the negative social impact after Conclusion section.

---

> ### Author Rebuttal · Authors · 2024-08-07
>
> _Thank you for taking the time to read our work. We are pleased that you find the paper easy to read, appreciate the residual nature of our model, and find the experiments sufficient. We address your concerns as follows:_
>
> - ___“...any design that guarantees that the output high-resolution frames are smooth over time?”:___
>
>     To encourage the temporal smoothness of high-resolution frames, our model employs temporal attention mechanisms in both the mean downscaling and residual diffusion modules. This smoothness is not explicitly hard-coded, but the fact that the input is a frame sequence and processed with temporal cross-attention enables the model to generate a temporally-coherent output sequence. This approach aligns with other video diffusion works [1] that achieve temporal coherence without hard-coding. For reference, note that we have provided sample output __videos in the supplementary zip__ (california.gif, himalaya.gif), which are as temporally smooth as the ground truth.
>
> - ___“...same conditions but different sampling noise, will output completely different results?”:___
>
>     Yes, as a conditional generative model, our approach produces different, stochastic high-resolution output sequences under the same input conditions. These samples differ in their high-resolution details but are all consistent with the low-resolution input. This variability is crucial in climate modeling as it captures the inherent uncertainty and variability in weather patterns, essential for generating ensemble forecasts. This is why we include CRPS as an evaluation metric since it helps us compare a set of stochastic samples against a deterministic ground truth. For better visualization, we have included __Figure 1 in the attached file in the global response section__. The figure depicts five stochastic samples that our model (STVD) generated for the same precipitation event in the Sierra Nevada region, as shown in the main paper. Additionally, we provide a variance map that depicts variability across these samples. Interestingly, we find that high variance correlates with high precipitation regions, reflecting the fact that precipitation is a highly stochastic and hard-to-predict phenomenon.
>
> - ___“...show experimentally that their approach is better at modeling extreme precipitation than traditional supervised methods?”:___
>
>     We agree that demonstrating our model's effectiveness in capturing the precipitation distribution, particularly extreme precipitation, is essential. In our paper, we address this through two metrics, Earth Mover Distance (EMD) and the 99.999th percentile error (PE), as highlighted in Table 1. These methods refer to the annual precipitation distribution, shown in Figure 4. Table 1 indicates that our method results in a low PE and EMD, indicating that we capture the annual precipitation distribution well and better than other methods. Figure 4 visually reveals that our approach also captures the tail end of this distribution better than other methods. This is what we mean by the statement that our approach is “better at modeling extreme precipitation”. We will clarify this in the paper.
>
> _Thank you again for taking the time to review our work. We hope that our clarifications and additional figures on stochastic samples address your concerns. We will incorporate the additional content prompted by your feedback in the paper revision._
>
> _[1] Ho, J., Salimans, T., Gritsenko, A., Chan, W., Norouzi, M., & Fleet, D. J. (2022). Video diffusion models. Advances in Neural Information Processing Systems, 35, 8633-8646._

---

> > ### Author Response · Authors · 2024-08-12
> >
> > Hi, we wanted to follow up on our recent rebuttal submission to ensure that our responses adequately addressed your concerns. If there are any remaining issues or if you have any further questions, we would be grateful for your feedback.
> >
> > Thank you for your time and consideration.

---

### Author Rebuttal · Authors · 2024-08-07

We thank the reviewers for taking the time to review our work and appreciate the detailed feedback and constructive comments. In this response, we address a common point about a claimed limitation of the paper. All other questions are addressed in reviewer-specific responses.

Some reviewers pointed out the use of “clean data” for training (e.g., the availability of paired high-resolution/low-resolution samples) as a potential practical limitation. We stress that this setup is an established way of performing precipitation downscaling, commonly called the 'perfect prediction' paradigm according to the taxonomy of [1]. In this approach, the coarse-grid features used for super-resolution are derived by coarse-graining the fine-grid data used for training. This most basic super-resolution task is an important first step toward a longer-term goal: training super-resolution models to enhance coarse-grid features from an inexact emulator of coarse-grid meteorology to super-resolve the fine-grid data. This process can be broken into two steps: (1) correcting biases in the emulator compared to the coarsened fine-grid data, and (2) performing the ‘perfect prediction’ task. The first step is emulator-specific, while the second is more generic and of broader interest.

In addition, some reviewer questions asked for additional investigations. We provide here a pdf containing the results of those investigations, containing

- Figure 1: Visualization of stochastic samples and variance map
- Figure 2: Visualization of temporal attention

Specific context and discussions are presented in the corresponding review rebuttals.

Many thanks,

The authors

_[1] Rampal, N., Hobeichi, S., Gibson, P. B., Baño-Medina, J., Abramowitz, G., Beucler, T., ... & Gutiérrez, J. M. (2024). Enhancing Regional Climate Downscaling through Advances in Machine Learning. Artificial Intelligence for the Earth Systems, 3(2), 230066._

---

### Decision · Program_Chairs · 2024-09-25

**Decision:**

Accept (poster)

**Comment:**

This paper presents a novel framework for precipitation super-resolution using a video diffusion model. It combines a deterministic downscaling module for initial results with a temporally-conditioned diffusion model to refine these results. The authors claim that this hybrid approach alleviates "mode averaging" issues and demonstrates effectiveness through experimental validation. Based on the three reviews, there is a general inclination toward acceptance, with no major flaws identified. I recommend accepting this paper as a poster. However, the authors should make further revisions before publication to meet publication standards. Specifically, the related work section lacks important references, the experimental details are unclear, and additional ablation studies are needed.